# Is *Hydra* Axis Definition a Fluctuation-Based Process Picking Up External Cues?

**DOI:** 10.3390/jdb13030024

**Published:** 2025-07-17

**Authors:** Mikhail A. Zhukovsky, Si-Eun Sung, Albrecht Ott

**Affiliations:** 1Institute of Endotypes in Oncology, Metabolism, and Immunology “G. Salvatore” (IEOMI), National Research Council (CNR), Via P. Castellino 111, 80131 Naples, Italy; m.zhukovsky@ieos.cnr.it; 2Biological Experimental Physics, Center for Biophysics, Faculty of Natural Sciences and Technology, Saarland University, Campus B2 1, 66123 Saarbrücken, Germany; sieun.sung@uni-saarland.de

**Keywords:** *Hydra*, *ks1*, *Wnt*, gene expression, Olami–Feder–Christensen model, power law, symmetry breaking, fractal, regeneration, spatial distribution, axis formation, temperature gradient, scale-free, avalanche-like dynamics, critical state

## Abstract

Axis definition plays a key role in the establishment of animal body plans, both in normal development and regeneration. The cnidarian *Hydra* can re-establish its simple body plan when regenerating from a random cell aggregate or a sufficiently small tissue fragment. At the beginning of regeneration, a hollow cellular spheroid forms, which then undergoes symmetry breaking and *de novo* body axis definition. In the past, we have published related work in a physics journal, which is difficult to read for scientists from other disciplines. Here, we review our work for readers not so familiar with this type of approach at a level that requires very little knowledge in mathematics. At the same time, we present a few aspects of *Hydra* biology that we believe to be linked to our work. These biological aspects may be of interest to physicists or members of related disciplines to better understand our approach. The proposed theoretical model is based on fluctuations of gene expression that are triggered by mechanical signaling, leading to increasingly large groups of cells acting in sync. With a single free parameter, the model quantitatively reproduces the experimentally observed expression pattern of the gene *ks1*, a marker for ‘head forming potential’. We observed that *Hydra* positions its axis as a function of a weak temperature gradient, but in a non-intuitive way. Supposing that a large fluctuation including *ks1* expression is locked to define the head position, the model reproduces this behavior as well—without further changes. We explain why we believe that the proposed fluctuation-based symmetry breaking process agrees well with recent experimental findings where actin filament organization or anisotropic mechanical stimulation act as axis-positioning events. The model suggests that the *Hydra* spheroid exhibits huge sensitivity to external perturbations that will eventually position the axis.

## 1. Introduction

Axis specification sets the spatial blueprint for tissue patterning and organogenesis, as well as the pathways of cellular differentiation during early development of multicellular animals. *Hydra* (Figure 1a) is a simple organism known for being immortal [1]. It can reform lost body parts. *Hydra* can even reorganize from a disordered cell aggregate into a fully formed structure, offering the opportunity to study how axis definition proceeds in the absence of pre-existing cues [2].

During such regeneration, *Hydra* first forms an isotropic, hollow cell spheroid that inflates due to osmotic swelling [3]. Similar lumen inflation patterns as in *Hydra* have recently been described in the early development of the mouse [4]. The authors identified pressure as a long-range mechanical signal that coordinates tissue self-organization across multiple scales. They suggested that “given the widespread presence of fluid-filled lumina in epithelial tissue morphogenesis (such as in the lung and kidney), it would seem natural to explore hydraulically mediated control of tissue size and cell fate in such systems”.

The role of mechanical stimulations in conjunction with the cytoskeleton during development has also been observed in other model organisms. For instance, in *Drosophila* embryogenesis, actomyosin contractions were shown to generate mechanical stress that regulates apical constriction and cell shape changes during the process of gastrulation [5,6]. Similarly, in *Xenopus laevis*, mechanical forces play a significant role in convergence and extension movements essential for gastrulation. Here, the cytoskeletal dynamics and cell–cell adhesion are regulated by the non-canonical *Wnt*/PCP signaling pathway, which is crucial for tissue elongation and axis formation [7]. In mammalian systems, mechanical forces transmitted via the cytoskeleton were shown to influence stem cell lineage specification. The YAP/TAZ signaling pathway responds to substrate stiffness and cytoskeletal tension, directing mesodermal differentiation [8,9]. These examples across various organisms illustrate the conserved role of mechanical forces and cytoskeletal interactions in regulating development and signaling pathways.

**Figure 1 jdb-13-00024-f001:**
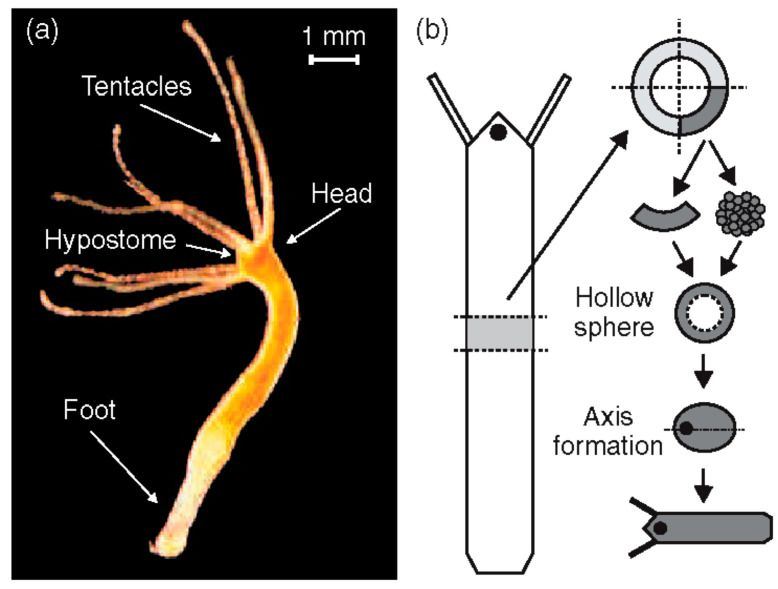
(**a**) Adult *Hydra vulgaris*; (**b**) sketch of the regeneration procedure. A ring is cut out of the middle of an adult, non-budding *Hydra*. The ring is cut into 4 pieces of equal size. These fragments of tissue close and form a hollow spheroid. The spheroid defines an axis first to form a complete animal of reduced size following the chosen direction. Reprinted, with permission, from Soriano et al. [10].

In the case of the *Hydra* spheroid, we suggest that besides the equally distributed long-range osmotic stimulus on the regenerating spheroid, there is only short-range communication limited to adjacent cells. We develop strong arguments that axis formation is due to collective fluctuations of expression patterns based solely on nearest neighbor communication [10]. These fluctuations are locked and the axis is defined as soon as the *Wnt* cascade is triggered, so that the early mouth diminishes the mechanical stimulation by releasing the pressure.

We also discuss our ideas in light of recent findings exploring the influence of mechanical stimulation and the structure of the actin cytoskeleton on the position of the future axis. We find that these observations integrate well into the suggested mechanism.

## 2. *Hydra* as a Model Organism

Cnidarians are evolutionarily old. Molecular evidence suggests that cnidarians appeared around 741 million years ago, before the Cambrian radiation, a period marked by the rapid emergence of most major animal phyla [11]. Their defining characteristic is the stinging cell, the nematocyte (also known as cnidocyte), which contains a unique toxin-producing organelle called nematocyst (also known as cnidocyst), used for predation and defense [12,13,14,15,16,17,18]. The phylum Cnidaria includes more than 9000 animal species, grouped into approximately seven classes and 25 orders.

*Hydra* (see Figure 1), a member of the phylum Cnidaria, class Hydrozoa, and order Hydroida, was first described in 1702 by Antonie van Leeuwenhoek and studied in more detail by Abraham Trembley [19]. The *Hydra* genome contains approximately 20,000 protein-coding genes, excluding transposable elements [20,21], and includes orthologues of genes known to play key developmental roles in bilaterian developmental models [20]. It was suggested that *Hydra* escapes senescence and may be potentially immortal [22,23,24,25,26,27]. *Hydra* can reproduce both sexually and asexually [28]. Under normal conditions, *Hydra* reproduces asexually by budding; however, it tends to reproduce sexually at the cost of becoming mortal if the environment becomes unfavorable [29]. Budding occurs about two-thirds of the way down the body column. This is where the body wall evaginates to form a new column that will develop a head at the end. This column eventually detaches as a new, small *Hydra* [30].

An adult *Hydra* is usually about a cm long. It consists of approximately 100,000 cells. All *Hydra* body parts are aligned along a single oral–aboral axis [31,32]. In the adult, new cells constantly emerge within the center column of *Hydra*, becoming terminally differentiated only when they migrate to either end, where cells are irreversibly lost. Differentiation is supported by four distinct stem cell lineages. Both ectodermal and endodermal epithelial cells are responsible for maintaining the epithelial bilayer by proliferating and migrating along the body column. As part of the tentacles and the foot, they become arrested in the G2 phase of the cell cycle [33,34]. They cannot form interstitial stem cells. These reside primarily in the ectoderm. They are multipotent and give rise to neurons, nematocytes, gland cells, and germline precursors. In turn, germline stem cells, derived from interstitial cells, contribute to gametogenesis and reproductive development [28]. This hierarchical organization of stem cells enables *Hydra* to efficiently replace lost tissues and to regenerate an entire organism from aggregates of dissociated cells.

*Hydra* has become a major model organism in development, among others, for its regeneration property [21,35,36,37,38,39,40,41,42,43,44,45,46,47,48,49]. A small fragment of tissue excised from an adult *Hydra* is sufficient to regenerate a complete organism within a few days (Figure 1b). *Hydra* can even reform from a heap of disordered cells [2]. In both situations, *Hydra* will first form a hollow cell ball. Since the body axis is lost in the disordered cell ball, approximately 10,000 cells must break the initial isotropy, define a single oral–aboral axis and organize as a function of it. Only cells from the body column contribute to regeneration, highlighting their multipotent properties [1,50].

Regeneration—the regrowth or repair of cells, tissues or body parts—is widespread but highly variable among animal phyla [51,52,53,54,55,56,57,58,59]. Similarities between regeneration and post-embryonic development suggest that regeneration may well have originated as a by-product, or epiphenomenon, of development [55,56,60], predominantly maintained by mechanisms other than direct selection [55]. According to the phylogenetic inertia hypothesis, the ability to regenerate is an ancestral trait retained for historical reasons; it is not selectively advantageous but persists because it has not been eliminated from the species’ developmental repertoire [55].

## 3. *Hydra* Genes Related to Axis Definition

Among the critical genes involved in early axis definition are β-catenin [61,62,63] and *Wnt* [62,64,65]. *Wnt* genes play axis-related roles in many animal species, including sponges. However, unicellular organisms do not contain these genes, suggesting that *Wnt* signaling played a crucial role in the origin of multicellular animals [66,67,68,69,70,71,72,73,74]. The *Wnt* gene family in *Hydra* is comparable in size to those in bilaterian animals [75], suggesting that the diversification of this gene family occurred before the cnidarian–bilaterian split [20].

*Hydra* expresses *Wnt* genes only in the organizer region. The organizer is part of the hypostome, located at the tip of the head (see Figure 1). The organizer is the first structure to be restored during regeneration [76]. It guides the fate of all other cells [77,78]. Vogg et al. [79] demonstrated that a conserved *Wnt* 3/β-catenin/Sp5 feedback loop restricts head organizer activity in *Hydra*, highlighting the role of *Wnt* 3 in axis formation. Studies have shown that disruptions in *Wnt* signaling pathways can lead to abnormalities in axis formation, underscoring their importance in *Hydra* development [61,64,65].

The complexity of *Wnt* intracellular signaling pathways in *Hydra* is reflected by the diversity of *Wnt* receptors [76]. These receptors can be categorized into 15 receptors and coreceptors from seven protein families [70]; the Fizzled family is the most prominent one [80].

β-catenin is located at the adherens junctions of the cell membrane prior to the activation of the *Wnt* pathway. The canonical *Wnt* pathway is characterized by the preservation of β-catenin from degradation in the cytoplasm. β-catenin eventually translocates into the nucleus, where it coactivates the TCF/LEF transcription factor family [81]. The non-canonical pathways include the planar cell polarity (PCP) pathway and the *Wnt*/calcium pathway, where alterations in gene expression are not mediated via β-catenin [81]. All three pathways, the canonical one, PCP, as well as *Wnt*/calcium, have roles in *Hydra* regeneration. For instance, treatment of *Hydra* cells with a GSK-3β inhibitor (alsterpaullone) leads to β-catenin translocation into the nucleus, causing head formation [82]. This translocation of β-catenin into the nucleus as a downstream step of the *Wnt* pathway has been extensively studied [83,84,85,86].

*ks1* is a *Hydra*-specific gene [87,88]. It is permanently expressed in the adult, just below the tentacles and hypostome, that is, before these cells differentiate irreversibly as part of the head. Its specific function remains unknown. Because *ks1* is upregulated in response to early signals of head formation, it has been termed a marker of “cell head forming potential” [89,90,91]. We monitored *ks1* in experiments described in the following sections.

## 4. *Hydra* Regeneration Requires Mechanical Stimulation

As a first step of *Hydra* regeneration, a hollow sphere made of a cell bilayer forms (Figure 1b). This does not depend on whether starting with a cell aggregate or a small piece of tissue. Subsequently, due to osmosis [3,92,93], cycles of steady inflation terminated by bursts are observed (Phase I, Figure 2a). The frequency and amplitude of these cycles abruptly change after about 18 h (Figure 2a). At this point, the release of pressure occurs at a decreased threshold without rupture and increased frequency (Phase II) as if a weak spot in the *Hydra* cell bilayer had emerged [3,10]. Futterer et al. [3] showed that this point in time goes along with a transition of the cell spheroid from elastic isotropy to anisotropy (Figure 2b). Moreover, the transition occurs around the time point where *Wnt* is supposed to be expressed, suggesting a potential connection between *Wnt* expression, cell adhesion and tissue elasticity [92]. Chiou and Collins [94] showed that indeed the emergence of an early mouth was responsible for the diminished pressure release threshold that induces the sudden change in the inflation burst cycles [92,95]. Soriano et al. [92] showed that axis definition is delayed by the same time interval as the inflation–burst cycles are prolonged due to changes in osmotic pressure, suggesting the osmotical inflation to be the driving force in the process. Duclut et al. [96] studied mechanical stimulation from osmotic pressure in cell spheroids in detail. Ferenc et al. [97] showed that mechanical oscillations orchestrate axial patterning through *Wnt* activation in *Hydra*. *Hy Wnt3* is more affected than other *Wnt* genes by the lack of spheroid inflation in an isotonic environment, indicating its sensitivity to mechanical cues. *Wnt* 3 overexpression in *Hydra* spheroids under isotonic conditions was able to rescue the regeneration process, highlighting its pivotal role in response to mechanical stimulation [97]. These observations point towards the role of the cell cytoskeleton as the force-sensitive structure of the cell.

**Figure 2 jdb-13-00024-f002:**
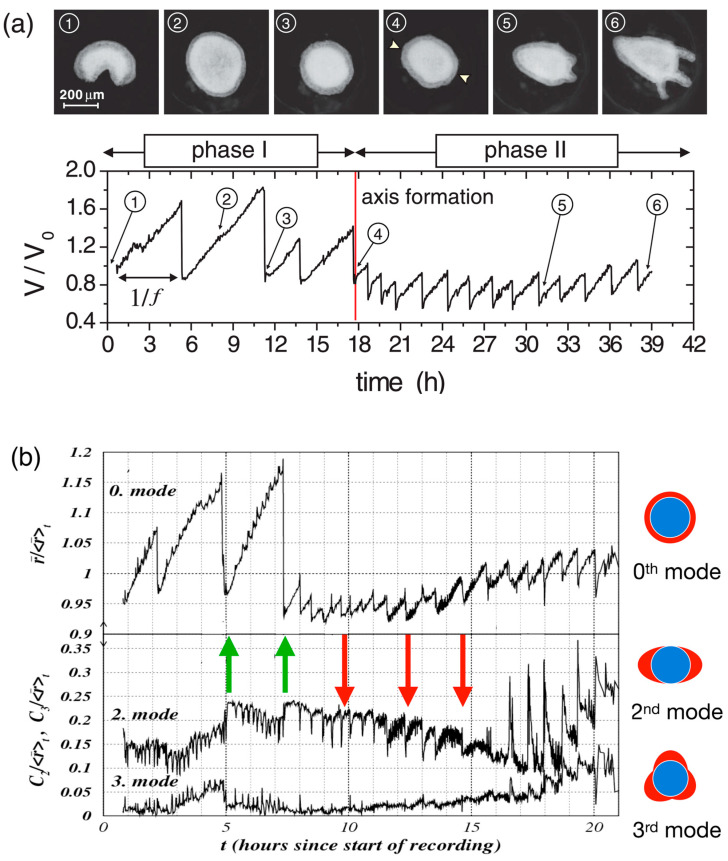
Mechanical oscillations during *Hydra* regeneration of a spheroid. (**a**) Volume as a function of observation time (below) with images (above) taken at time points as identified. We call phase I the high amplitude, low frequency inflation burst cycles, and the following low amplitude and high frequency oscillations phase II. Reprinted, with permission, from Soriano et al. [92]. (**b**) Temporal evolution of shape deformation modes by Fourier transform during *Hydra* spheroid regeneration. The 0th mode represents isotropic size changes (the average radius), while higher modes (2nd and 3rd) capture anisotropic deformations as shown. Green arrows mark instants where the second mode increases as a result of sudden pressure release; red arrows mark instants where the second mode decreases as a result of sudden pressure release. It can be easily spotted that in phase II, the 0th and the 2nd mode go increasingly in sync. Note that the behavior following the green arrows corresponds to elastic isotropy of the tissue, while the behavior of the red arrows corresponds to elastic anisotropy (the spheroid becomes more oblong upon inflation). We understand that after early mouth formation, the tissue changes its elastic properties following an axis that has been set. Reprinted, with permission, from Fütterer et al. [3]—modified by adding colored arrows and colored pictures concerning the modes of deformation.

## 5. Symmetry Breaking in Physics and Biology

Symmetry breaking in non-equilibrium has been studied in Physics at length [98,99,100]. To make a transition to asymmetry, these systems tend to generate transient asymmetries through fluctuations. This is still a symmetrical situation. Only if the asymmetric fluctuation reaches a critical size so that it becomes locked is the symmetry considered broken. As an example, consider a vortex as it may appear in a sink. Initially, when the liquid starts to drain, there will be no vortex, but increasingly large fluctuations due to the large flow of liquid create transient asymmetry. Only when these fluctuations reach a sufficient (critical) amplitude to become self-amplifying, the symmetry is broken and the vortex will increase in strength. At that point, the vortex will not reverse course if left to itself. It will resist increasingly large perturbations in the opposite direction, even if these would have been large enough to impose a direction if they had been applied from the start. In other words, these perturbations can be larger than the initial fluctuation spectrum and still have no effect at this point. At the same time, by very violently spinning the entire mass of fluid in the opposite direction, the direction of any vortex can be changed—but this does not mean that the vortex did not undergo symmetry breaking earlier, or that there was no clearly defined direction. We understand that for a study, the magnitude of the applied perturbations matters a lot.

If we ask, how we can show that the vortex broke symmetry in the beginning, it becomes clear that two features are important: (i) initially the direction of rotation will be imposed by perturbations that create rotational flow at the critical fluctuation level—this is still very small compared to an established vortex; (ii) once a direction is chosen, there must be an amplification and stabilization of the movement (Chapter 10.2 “Critical Precursory Fluctuations” in Sornette [101]).

The same idea is found in Biology. Regarding external influences on developmental decision, “*Molecular Biology of the Cell*” by Alberts et al. [102] says (Chapter 22 “Development of Multicellular Organisms”): “The choice between the alternative outcomes can be dictated by an external signal that gives one of the cells a small initial advantage. But once the positive feedback has completed its work, this external signal becomes irrelevant. Broken symmetry, once established, is very hard to reverse: positive feedback makes the chosen asymmetric state self-sustaining, even after the biasing signal has disappeared. In this way, positive feedback provides the system with a *memory* of past signals”.

Transferred to *Hydra*, this means that we have to show that once a direction is chosen, it amplifies (and, as discussed in Section 11, this seems to be the case). However, the property that small perturbations or advantages are sufficient to direct the axis remains largely unexplored, except for what follows.

## 6. A Weak Temperature Gradient Can Direct the Future Axis of a Regenerating *Hydra* Spheroid

The above can be applied to developing *Hydra* spheroids that are exposed to a temperature gradient. If the temperature gradient of typically less than a degree was applied early, during phase I (high amplitude, low frequency oscillations), it would set the axis. However, if applied during phase II (low amplitude, high frequency oscillations), the gradient could no longer change the direction of the axis. Soriano et al. [10] concluded that the axis direction of the regenerating *Hydra* spheroid becomes irreversibly locked during the transition from phase I to phase II (Figure 2). Larger fragments that retain a certain degree of asymmetry were found to switch to phase II much faster [92].

Naively, one might expect the head to appear on the warmer side, given that the organizer is the first structure to emerge during regeneration. This is because the increased temperature accelerates the development, and this creates the ‘small advantage’ (see above) that can be expected to propagate to axis positioning. However, in cases where the temperature difference between the cold and hot sides was ΔT = 0.6 °C, the head formed on the cold or the hot side with about equal probability (Figure 3). However, if the gradient was increased to ΔT = 0.9 °C, a tendency for the head to form on the cold side became apparent [103]. Not only did the direction of a temperature gradient impose the position of the axis, but the magnitude of the gradient had an effect on the orientation, not an intuitive result.

**Figure 3 jdb-13-00024-f003:**
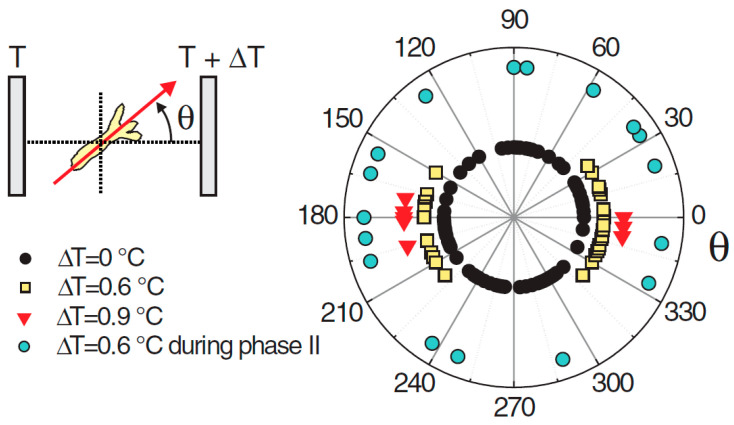
Angular distribution of the *Hydra* head orientation with respect to the direction of a temperature gradient. The temperature difference ΔT across the spheroid is an estimation from the temperature set on the sapphire plates and the distance between them. Reprinted, with permission, from Soriano et al. [10].

## 7. Ks1 Expression Patterns at the Axis Defining Moment Are Devoid of a Characteristic Size

Usually, we define our surroundings using specific scales, such as the typical size of organisms or their typical lifespans. However, some phenomena do not fit into this kind of description. Since they lack a characteristic size or scale, they are called scale-free. Mathematically, the absence of scale corresponds to a power law, which is surprisingly common in nature [104]. A power law can be expressed as:*f*(*x*) = *b*⋅*x^a^*

Here, *x* is the input, for instance, the scale of a map, and *f*(*x*) is the output, for instance, the length of the coastline of Britain, taking a map at that scale. If the exponent *a* is one, the two quantities are just proportional. An exponent different from one means that one grows faster or slower than the other. Doubling the scale of the map does not necessarily double the length of the coastline because of its rough, intricate shape. “*a*” is the exponent of the power law that defines the quantitative connection between the input, the scale of the map, and the output, here the length of the coast. *a* is called the fractal dimension. More precisely, *a* is called the “perimeter fractal dimension” of the coastline of Britain because it yields the corresponding length. Its value is estimated to be around 1.25 according to Mandelbrot’s original analysis [105].

One key feature of a power law is that if you scale the input *x*_0_ by a factor of *k*, the output of the function *f*(*k*⋅*x*_0_) will be increased by *k*^a^ with respect to *f*(*x*_0_):*f*(*k*⋅*x*_0_) *= b*⋅(*k*⋅x_0_)^a^ = *b*⋅*k*^a^⋅x_0_^a^ = *k*^a^⋅*f*(*x*_0_).

Since this applies to any *x*_0_, regardless of its precise value, the function *f*(*x*) is scale-free: at whatever scale you look at the function, its shape remains the same. Scale-free can be shown to indeed be equivalent to a power-law description [104].

Non-trivial scale-free objects are known as fractals. The term “fractal” was first used by Benoît Mandelbrot in 1975. He derived it from the Latin word “frāctus”, meaning broken or irregular. Fractal analysis was shown to be useful in the biological sciences, helping to describe complex patterns and structures in nature [106,107,108,109,110,111,112,113,114], including investigation of cnidarians [115,116]. Fractal properties of gene expression were reported by Manoel et al. [117], Ghorbani et al. [118], and Waliszewski [119].

Fractals are fragmented geometric shapes that can be broken down into smaller parts, each of which is a reduced copy of the whole [113,114,115]. If you take photos with different magnifications of the same fractal, these photos will, again, look strictly the same. An example of this in nature is Romanesco broccoli (Figure 4), where each floret looks like a miniature version of the whole vegetable.

**Figure 4 jdb-13-00024-f004:**
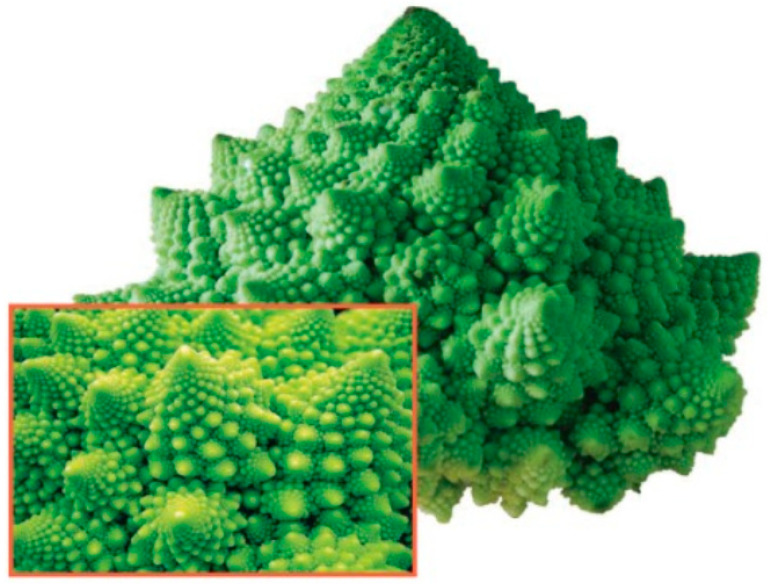
The Romanesco broccoli, a natural approximation of a fractal. Reprinted, with permission, from Bancaud et al. [120].

If you cover a straight line with a grid of square boxes (imagine pixels of a camera), the number of boxes *N*(δ) required is inversely proportional to the box size δ. Mathematically, this is expressed as *N*(δ) = constant/δ, leading to a slope of 1 when plotting ln(*N*(δ)) against (−ln(δ)) for a straight line. The perimeter fractal dimension in this case equals 1. Indeed, a line is a one-dimensional object. A more complex line shape, such as the coast of Britain, however, will most likely lead to a different slope when plotting the logarithms. The slope represents the fractal dimension (Figure 5).

For a simple, two-dimensional shape like a square, the area fractal dimension would be 2, indicating that it fully occupies a flat geometric plane, corresponding to the geometric dimension of flat shapes. This is because the number of boxes needed, *N*(δ), is inversely proportional to the area of the box δ^2^. This leads to a slope of 2 in the plot of ln(*N*(δ)) versus −ln(δ). For a more complex shape, again, the fractal dimension is likely to be different from 2.

In the study by Soriano et al. [10], the spatial distribution of *ks1* gene expression on the surface of the *Hydra* was examined using in situ hybridization. There were domains (or clusters) of cells expressing the *ks1* gene. These domains were analyzed to check if they were fractal objects. This was carried out using the grid (or box-counting) technique as outlined above (see Figure 5).

The perimeter fractal dimension *D*_per_ of the *ks1*-expressing domains was determined for three culture times [92]. The results are given in Figure 6. The fractal dimensions reached a maximum above 1.3 after about 25 h, which suggests that the spatial distribution of *ks1* expression exhibits fractal properties throughout the critical period of symmetry-breaking and axis definition. Similarly, the area fractal dimension *D*_area_ was determined as 1.7, using the grid method [121].

Besides studying the shape of the *ks1*-expressing domains, Soriano et al. [10] also studied their size distribution. A domain size *s* means that it contains *s* cells. To diminish the noise in the analysis, for each value of *s*, the probability *P*(*s*) of observing a domain with a size equal to or larger than *s* was extracted from the data and plotted as a function of *s* (Figure 7). Around the transition to phase II, the point from which the axis appears locked in a temperature gradient (at a culture time of 25 h), the relationship between *P*(*s*) and *s* appears linear on a log-log scale. The slope of this relationship corresponds to the scaling exponent −τ, which is close to −0.8. Therefore, the probability *P*(*s*) as a function of domain size *s* can be described by a power law *P*(*s*)∼*s^−^^τ^*. Since the domains scale in size as well as in their shape, which is fractal, one can say that the spatial gene expression pattern of *ks1* on the spheroid becomes scale-free at the axis locking moment. For the other culture times (16 h and 48 h), the distribution does not fit a power law: here, characteristic scales are present in *P*(s). The observation of scale-free properties often allows one to isolate possible underlying mechanisms because they are unusual. We present a hypothetical model that considers that in the following figure.

**Figure 5 jdb-13-00024-f005:**
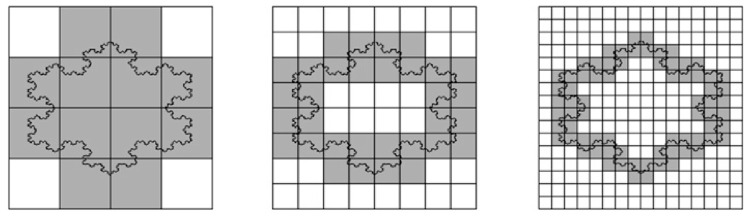
Box-counting method for the calculation of perimeter fractal dimension. Boxes decrease in size by a factor of two between neighboring images (from left to right). A straight vertical line would require a proportional increase in box number (4, 8 and 16), leading to a fractal dimension of one. However, the irregular perimeter here requires 12, 32, and 72 boxes, respectively, indicating a deviation from linear scaling. If we obtain a straight line on a log-log plot (of the number of boxes as a function of grid size), fractal dimension is given by its slope. Reprinted, with permission, from Florindo and Bruno [122].

**Figure 6 jdb-13-00024-f006:**
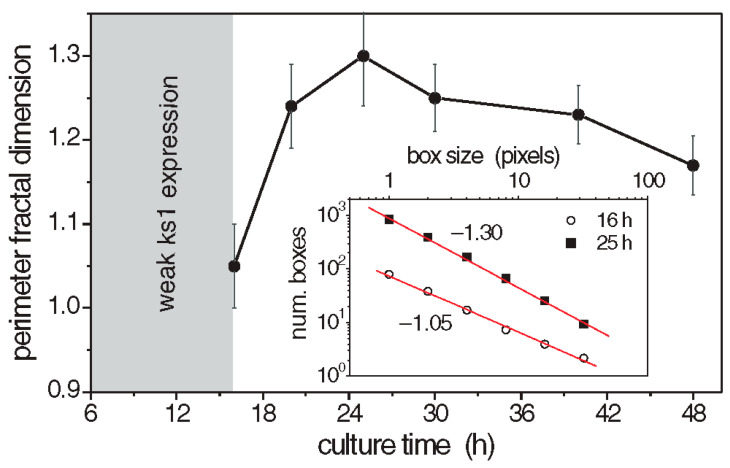
Perimeter fractal dimension of *ks1* expression as a function of time. Each point represents the average of 6 analyzed patterns. Inset: Examples of log-log fits to determine the perimeter fractal dimension of two studied patterns. Reprinted, with permission, from Soriano et al. [10].

**Figure 7 jdb-13-00024-f007:**
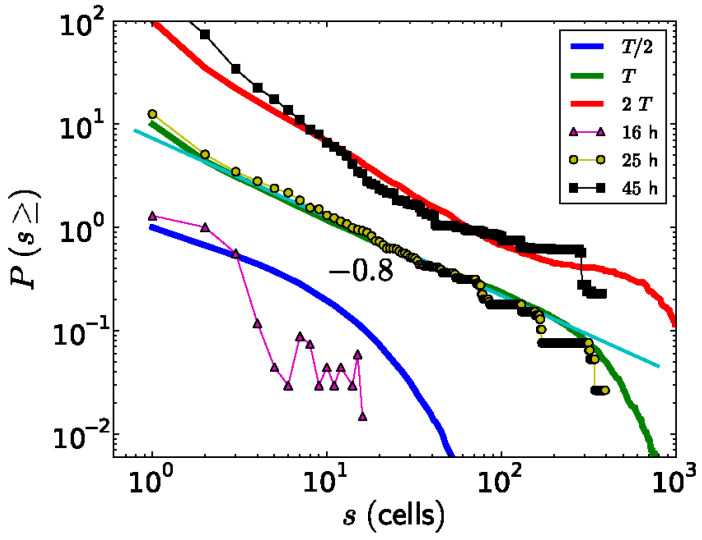
The probabilities *P(s)* to experimentally observe a *ks1*-expressing domain with size equal or larger than s, for three different culture times (dots, the thin lines represent guides for the eye). This is compared to the simulated size distribution of avalanches appearing at three time points, fat curves (for a conservation level of *C* = 0.95). For better visualization, the curves were shifted vertically. The straight thin, light green line is a power-law fit to the experimental data, yielding an exponent of −0.8, equal to the result from the simulation. Reprinted, with permission, from Gamba et al. [103].

## 8. Model Based on Production and Exchange of Ks1-Promoting Factor

Cells communicate with each other through different types of signaling, such as endocrine, paracrine, and juxtacrine communication [102]. In the case of *Hydra*, there is currently no evidence for endocrine signaling during axis definition, which raises the question of how the *Wnt*-expressing organizer communicates with other regenerating cells within the organism. Paracrine signaling is well-documented in other animal developmental processes [123,124,125,126,127,128,129] as well as plant developmental processes [130,131]. Paracrine factors, signaling molecules that diffuse over short distances, can regulate gene expression in neighboring cells [132,133].

Epithelial cells are often connected by gap junctions, microscopic channels that enable the transport and communication of signaling molecules between adjacent cells [134,135,136]. In *Hydra*, the cells in the body column are extensively connected by these gap junctions, facilitating intercellular communication. Their role in intercellular signaling suggests that they could contribute to nearest neighbor signaling during axis definition [137].

Cellular functions often arise from the interactions among neighboring cells, leading to synchronized responses [138,139]. This aligns well with the formation of *ks1* expression domains on the *Hydra* cell spheroids. *ks1* expression is either fully turned on or off, resulting in synchronized *ks1* expression among neighboring cells that leads to the observed domains.

Nearest neighbor signaling in only one dimension will always lead to exponential decay in that direction. In two dimensions, however, the signaling may find new directions so that it spreads exponentially with increasing probability over time. As a result, there is a chance that the result obeys a power law, becoming scale-free. Nevertheless, the probability of signaling to the nearest neighbor needs to be tuned to a precise value for that to happen. Self-organized critical models tune to that parameter by themselves. This often occurs in a robust way, meaning that the result is only slightly changed by reasonable changes in the interaction between neighbors [104]. We, therefore, searched for a self-critical model that could reflect the situation in *Hydra*.

The model relies on the production of a hypothetical *ks1*-promoting factor, “*X*”, which is transferred between directly neighboring cells [103]. The exact contributors to *X* remain undefined; however, the concentration of *X* in a cell integrates various signaling inputs, including paracrine factors or other elements in the surrounding environment, affecting the likelihood of *ks1* expression of a given cell. β-catenin, which must be released from the cell junctions into the cell nuclei under the mechanical stretch from the osmotic inflation, may well be the most important contributor to *X*. Initially, each cell has a concentration of *X* that is attributed at random between zero and just below *[X]_c_*. Since the osmotic inflation slowly drives the cells to increasingly express *ks1*, we impose that *X* increases slowly but steadily at a rate *ν = d[X]/dt*, which means that in the simulation the *[X]_i_* are increased one by one, each by adding a small amount in random order. As *[X]_i_* in the *i*-th cell reaches *[X]_c_*, a determined fraction of *[X]_i_* is distributed evenly to its six neighboring cells. *[X]_i_* is set to zero and *ks1* is expressed by the i-th cell. As *X* is shared among neighboring cells, the concentration in these cells may also reach the threshold, in turn initiating expression. This process creates a chain reaction, or avalanche, of *ks1* expression that spreads across cells, forming a *ks1* expression domain (Figure 8a).

The larger the avalanche, the larger the domain of *ks1*-expressing cells formed by this process. Once an avalanche ends, the *ks1* expression domain disappears. Eventually, a new avalanche begins, usually at a different location on the sphere. Throughout the simulation, *ks1* expression in each cell starts and stops multiple times. For the model to function correctly, the avalanche spreading rate must be much faster than the rate of production of the *ks1*-promoting factor *X* [103]. This makes the *ks1* expression pattern evolve slowly, leading to progressively larger avalanches. The result is stable if a certain degree of randomness in *X* production across cells introduces variability in avalanche initiation, reflecting “natural” fluctuations in *ks1* expression.

**Figure 8 jdb-13-00024-f008:**
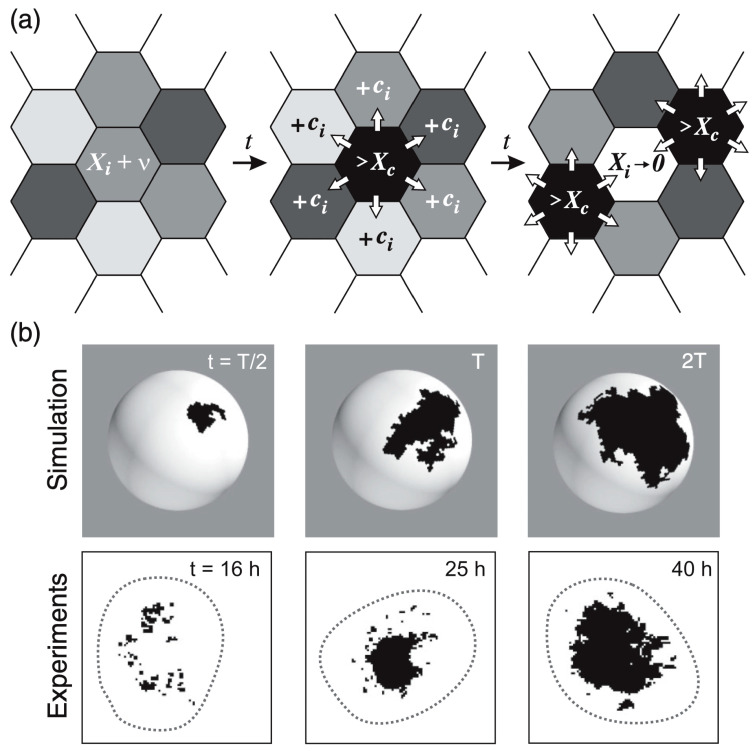
Avalanches of *ks1* activation in *Hydra*. (**a**) In this model, each cell produces a hypothetical *ks1*-promoting factor *X* over time. When *X* exceeds a threshold *[X]_c_* in a cell, the cell expresses *ks1*, resets its *X* level to zero, and distributes 95% (given by the conservation level C, here 95%) of its *X* equally to six neighbors. This may trigger a cascade (“avalanche”) of *ks1* activation if neighboring cells also reach the threshold. (**b**) Comparison of *ks1* expression (black spots) in numerical simulations and experiment, at three different development times. T is the simulation time associated with the emergence of a power law *P(s)*~*s^−τ^*, which takes approximately 25 h in the experiment. Reprinted, with permission, from Gamba et al. [103].

## 9. Analysis of the Model: Results of Simulations

In the study by Gamba et al. [103], numerical simulations were carried out to test the proposed model for *ks1* expression. Although the body wall of *Hydra* consists of two cell layers (ectoderm and endoderm), *ks1* is expressed exclusively in the ectodermal (outer) layer [91]. Therefore, the simulations modeled the *Hydra* ball as a single-layered structure representing only the ectoderm. This simplification was justified by additional analyses showing that introducing a second layer did not significantly alter the overall model dynamics [103]. A *Hydra* spheroid was modeled as a spherical lattice of approximately 10,000 hexagons, each representing a cell (see Figure 8a). The simulations showed that the best match to experimental data occurred if 95% of *X* was conserved during transfer to neighboring cells.

Following the rules outlined above, only small avalanches of *X*-releasing cells were observed initially. On the sphere, they appeared at random locations. Over time, the domains increased in size, and after approximately 2000 avalanches, the system reached almost a steady state where domains of all sizes were observed.

Examples of typical domains of cells expressing *ks1*, emerging from the avalanche mechanism of the model, are shown in Figure 8b. The three domains in Figure 8b represent independent events: each domain emerged and disappeared before the next appeared. To make them appear at the same time, one would have to keep the black color for longer. This would not affect the results in any way.

The simulated expression domains were described by examining the distribution of avalanche sizes, *P(s)*, which represents the probability of observing an avalanche (or domain) of *ks1*-expressing cells larger than size *s*. Initially, *P(s)* displayed a more or less exponential decay. Over time, the distribution evolved to a power-law, indicating that the size distribution of the *ks1* expression domains became scale-free. Moreover, at this point (time T), the expression domains formed fractal-like patterns. The model yielded a perimeter fractal dimension *D_per_* ≈ 1.28 and an area fractal dimension *D_area_* ≈ 1.75. The power law of *P(s)* disappeared for longer times (2T).

## 10. Comparison of Numerical Simulations with Experimental Data

Figure 8b compares typical shapes of the main sizes. Figure 7 shows the size distribution of avalanches (corresponding to the domains of *ks1*-expressing cells) that appeared at simulation times T/2, T and 2T, where T denotes the time until emergence of a power law *P(s)*~*s^−τ^*). These results were compared to the equivalent experimentally observed distribution, *P(s)*, of *ks1*-expressing domains (as described above) for the three culture times of 16 h, 25 h, and 45 h. A strong correspondence appeared between the numerical simulations and experimental data: for both, the linearity of ln(*P(s)*) vs. ln(s) extends over two orders of magnitude (decades), with an identical exponent of τ ≈ 0.8.

The values of perimeter fractal dimension and area fractal dimension (*D_per_* = 1.28 ± 0.05 and *D_area_* = 1.75 ± 0.05) obtained from numerical simulations for the conservation level C~0.95 at simulation time T closely match the experimental data, where the perimeter fractal dimension is approximately *D_per_*~1.3 at 25 h and the area fractal dimension is *D_area_*~1.7. This similarity indicates that the time T associated with the emergence of the power law *P(s)*~*s^−τ^* in the simulations is likely close to the experimental time t ≈ 25 h of the *Hydra* axis formation.

As described above, experiments demonstrated that a temperature gradient could direct the position of the emerging axis if applied sufficiently early. Increasing temperature corresponds to accelerating the rates, which is simple to do in the simulation. Usually, the acceleration in rates as a function of temperature in biology follows a Boltzmann description [140,141]. which was realized in the simulation. Supposing that an overcritical, large avalanche of *ks*1 expression, above a certain threshold, is locked to define the axis, the head emerged at the cold and hot sides of the spheroid with about equal probability [103]. However, the head had an increased tendency to appear at the cold side for larger temperature gradients. This corresponds to what was observed experimentally.

## 11. Mechanical Stimulation and the Actin Cytoskeleton

Since our initial work regarding osmotic inflation and burst cycles of a regenerating *Hydra* spheroid [3], quite an amount of work has focused on the connections between mechanical stimulation, *Hydra* axis definition and *Wnt* expression.

Adult *Hydra* exhibits filamentous actin, oriented in concentric rings in the endoderm and in the direction of the axis in the ectoderm [142]. Braun and Keren [49] found that asymmetry and defects in nematic actin fiber orientation of the cell spheroid correlate with the position of the future axis. Ravichandran et al. [143] demonstrated that mechanical confinement can induce a second axis in regenerating *Hydra*. Their study showed that in toroidal *Hydra* tissues, regeneration occurred only if actin cytoskeletal defects were present, directly implicating such defects as necessary triggers for axis formation. This provided clear experimental evidence that topological or mechanical defects can act as organizers of morphogenesis. In a different setting, Maroudas-Sacks et al. [144] regenerated *Hydra* spheroids with inherited axes inside narrow tubes. When the inherited axis was misaligned with the tube axis, polyps sometimes developed two heads instead of one head and one foot. This was interpreted as a reversal or loss of polarity, rather than true axis duplication, and was linked to mechanical constraints and the emergence of multiple defects in actin fiber orientation. However, such polarity reversals appeared to require substantial perturbation.

Through exactly which pathway actin is linked to directional signaling in *Hydra* remains elusive. However, actin is connected to the cell junctions, where β-catenin is located. Moreover, actin is the force-generating structure of cells. Active contraction of cells occurs in the direction of actin [145]. It may well be that the increased stretch of the tissue occurs predominantly at the locations where a defect is located, leading to the result above [143,144,146]. At the same time, the situation may be more complex. Microtubules are connected to actin via the Rho signaling pathway [147]. Both elements are known to act in concert [148]. Often, microtubules generate directional information in cells during development [149,150,151,152,153]. We showed that *Hydra* regeneration is prevented by nocodazole, unless it is rescued by exchanging nocodazole for paclitaxel. The delay in regeneration corresponds to the moment of rescue [154].

Asymmetric mechanical constraints can set the axis [148,149]. The feedback between stretch-induced *Wnt* expression, β-catenin nuclear presence and decreased tissue stiffness has been clearly evidenced [97,155]. During osmotic inflation, this is likely to produce instabilities where elastic weakness entails faster stretching, meaning increased probability of *Wnt* expression, in turn translating to increased elastic weakness. This mechanism successfully explained the experimental data in Weevers et al. [155] without the need to consider defects in the actin filament network. Bailles et al. [156] studied regeneration from single cells where supracellular actin organization was absent at the beginning. The actin fibers gradually self-organized; however, it was not clear whether this structural emergence was functionally relevant in terms of axis definition. Actin organization could not be linked to the emergence of *Wnt* expression.

One may ask how all the data on mechanical signaling can be linked to the model that we presented earlier. If mechanical stretching increases the probability of *Wnt* expression and head formation (and by extension, of *ks1* expression), this simply means increased production of the *X* factor in our model. Obviously, large, avalanche-like, synchronized expression patterns would preferentially appear at the location of increased production. Following this interpretation, stimulation through mechanics means that the imposed stimulus is substantially above the natural fluctuation level, so that it will eventually direct the symmetry-breaking process. Compared to the analogy of vortex formation, this means that the liquid is actively stirred in a given direction, above the natural fluctuation level, and before vortex formation, so that the emerging vortex just follows this direction.

In the shape analysis of a *Hydra* spheroid during regeneration [3,146], we did not detect elastic tissue anisotropy before the appearance of the early mouth, in excellent agreement with Bailles et al. [156]. However, this does not mean that externally applied anisotropic stretch will not translate to asymmetry before that time point. Following our interpretation, it just means that the system does not have to generate a large fluctuation of the *X* factor by itself that will break the symmetry.

The OFC (Olami–Feder–Christensen) model is part of the so-called self-critical models. They evolve towards a critical state on an infinite lattice. By definition, these states are not only scale-free, but they also provide infinite sensitivity towards perturbation [104]. This is because of the absence of characteristic scales. Although *Hydra* is finite, some of the sensitivity can be expected to remain present. We believe this explains the response to the small temperature gradient, as well as the response to structural defects of the actin cytoskeleton. It would make sense if this ‘non-molecular’ way of symmetry breaking were a vestige from the transition to multicellular animals, where the associated molecular pathways were poorly developed.

## 12. Conclusions

We discussed results on the expression of the ‘marker of head forming potential’ *ks1* [10] and a proposed theoretical model [103] that reproduced the results quantitatively. A temperature gradient as a non-molecular parameter changes the biochemical rates on both sides of a regenerating spheroid, leading to differential aging. Supposing that an overcritical (sufficiently strong to break the symmetry) fluctuation is locked to break symmetry, the intricate behavior of *Hydra* axis positioning as the animal is exposed to a (decidedly weak!) temperature gradient is predicted by the model. Quoting von Neumann’s saying, “with four parameters I can fit an elephant, and with five I can make him wiggle his trunk” [157], one cannot stress enough that this model quantitatively produces all the results with only a single adjustable parameter. We suggest that the gene expression patterns of the cells that form the regenerating *Hydra* sphere increasingly synchronize until a supercritical fluctuation, that is, a fluctuation that eventually leads to stable *Wnt* expression, emerges. This interpretation is supported by observations that, during axis formation, *Wnt*-expressing cells can emerge in very limited numbers, too few to give rise to a stable organizer. We interpret this phenomenon as the result of subcritical fluctuations of insufficient magnitude to break symmetry [158,159].

We show how mechanical perturbations that increase the probability of *Wnt* expression naturally become part of the theoretical model; however, we argue that mechanics is likely to be a strong perturbation overriding the natural fluctuations in an unperturbed setting. A competition between at least two different stimuli that can be varied in magnitude and may offer a better view of the contributions to the symmetry-breaking process in the future. Moreover, if other, unrelated small stimuli are applied, this may offer a lever to test the proposed theoretical model, since *Hydra* is predicted to position its axis as a function of almost anything.

## Data Availability

No new data were generated in this review. All referenced data are from previously published studies (see citations), and permissions for reuse have been secured from the original copyright holders.

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
