# Peer review of "Is Hydra Axis Definition a Fluctuation-Based Process Picking Up External Cues?"

_jdb, 2025, doi:10.3390/jdb13030024_

Round 1
Reviewer 1 Report
Comments and Suggestions for Authors
The review by Zhukovsky, Sung and Ott highlights recent and older studies on the mechanism of patterning, symmetry breaking and morphogenesis in the Hydra, highlighting that some concepts and quantitive descriptions that were probably not given the deserved credit at the time of their publication now prove their importance under more recent findings.
Overall, the review is well written, physical concepts being explained to the lay public of biologists. I therefore have only minor comments:
section 4 (Hydra genes related to axis definition): The authors mention that plant do not have a Wnt signalling pathway to set the axis, which is true. The authors should however mention that plants have Auxin, which works very similarly to Wnt, forming a gradient from the tip, and participating the creation of new axes (branches and leaves). It also has a mechanical feedback loop like Wnt, and it secretion source is correlated with a nematic defect in the orientational order of cells and microbutules. Please see the work of Olivier Hamant, Jan Traas and Teva Vernoux to cite the least.
section 6, first sentence, "in physics" is repeated two times in the same sentence.
section 9: paragraph 3: correct sentence "...there is a chance that the result obeys to a power law, becomING scale free."
in section 12: paragraph 12: attribution of discoveries is inappropriately reported between citations 154 and 155. 155 showed first that mechanical confinement could create a second axis (155 was published after 154, but 155 had been on bioRXiv a few month before 154). Also, 154 does not show axis duplication, but rather a polarity loss or reversion, the hydra forming 2 heads instead of a foot and a head. We understand that the authors want to link this axis reversion or polarity loss to their mode, but the distinction between the citations should be made clearly. Finally The absolute proof that defects are required to head formation comes from the toroid experiment, as said by the authors, bu this should be clearly said.
Reviewer 2 Report
Comments and Suggestions for Authors
This paper is interesting and useful for improving our understanding of the role of mechanical forces in development and regeneration. It will be of interest to developmental biologists, embryologists and zoologists. It is written in simple, clear language.
However, I recommend that the authors make some corrections to the text.
1) The abstract begins with the sentence, 'Axis definition is of primordial importance for embryogenesis'. However, neither the abstract nor the article goes on to discuss embryogenesis. I recommend removing or replacing this sentence.
2) I recommend that the authors significantly revise the first four sections: 'Introduction', 'Hydra: an overview', 'Hydra regeneration' and 'Hydra genes related to axis definition'.
The information in these sections is poorly structured, and the order in which it is presented is not well thought out. These sections could be considerably shortened. For example, at the end of the section 'Hydra: an overview', data from rather old works are given which are not relevant to the subject of the article. Without additional explanations, they only cause confusion. ("Consequently, the axial organization of Hydra provides important insights into the transition from radial to bilateral body plans and the evolution of the bilaterian body plan [44-46]. The Hydra foot, which pumps gastrovascular fluid, and the blood-pumping vertebrate heart were proposed to have a common ancestry [49], with the foot-to-head extension in Hydra corresponding to the anteriorposterior axis in higher organisms [50].)
Although the authors claim that the article is intended for biologists, the biological information is presented in a highly simplified manner. For example, the mesoglea is described as a 'noncellular, jelly-like substance'. The terms 'stem cells' and 'i-cells' are not used anywhere. The connection between the evolution of the Wnt signalling cascade and the origin of Metazoa ('Hydra genes related to axis definition') is overly simplified.
It seems to me that the authors should decide whether the article is intended for physicists (or mathematicians) with limited knowledge of biology or professional biologists.
The other sections of the article are really well written, and I have no comments on them.
Round 2
Reviewer 1 Report
Comments and Suggestions for Authors
The authors have fully answered my concerns and I enthusiastically recommend publication.
Author Response
Comment 1: The authors have fully answered my concerns and I enthusiastically recommend publication.
We thank the reviewer for his valuable comments and his enthusiasm.
Reviewer 2 Report
Comments and Suggestions for Authors
The article has improved significantly compared to the original version. My primary concerns were related to the structural weaknesses of the paper, which the authors have successfully addressed. However, I still find the first part of the abstract unsatisfactory. I recommend replacing it with the version I suggest here or a similar alternative written by the authors themselves.
The author's version: Axis definition often plays a key role in tissue pa4erning. Hydra has extraordinary regenerative abilities. During regeneration of Hydra from a random cell aggregate or a sufficiently small fragment of tissue, a hollow cell spheroid emerges that undergoes symmetry breaking and de novo axis definition.
My suggestion: Axis definition plays a key role in the establishment of animal body plans, both in normal development and regeneration. The cnidarian Hydra can re-establish its simple body plan when regenerating from a random cell aggregate or a sufficiently small tissue fragment. At the beginning of regeneration, a hollow cellular spheroid forms, which then undergoes symmetry breaking and de novo body axis definition.
Author Response
Comment 1: The article has improved significantly compared to the original version. My primary concerns were related to the structural weaknesses of the paper, which the authors have successfully addressed. However, I still find the first part of the abstract unsatisfactory. I recommend replacing it with the version I suggest here or a similar alternative written by the authors themselves.
Answer: We changed the abstract according to the reviewer's suggestions. We thank him for the improvements and for all the work he put into our manuscript.